# Green Microwave-Assisted Synthesis of CoFeRu-Based Electrocatalyst for the Oxygen Reduction Reaction

**DOI:** 10.3390/ma14071662

**Published:** 2021-03-28

**Authors:** Antonia Sandoval, Edgar Borja, Lorena Magallón, Javier Su

**Affiliations:** 1Cátedras-CONACYT-Centro de Investigación y Desarrollo Tecnológico en Electroquímica S.C., Querétaro 76703, Mexico; asandoval@cideteq.mx; 2Departamento de Física y Química Teórica, Facultad de Química, Universidad Nacional Autónoma de México, Ciudad de México 04510, Mexico; javiersugallegos@gmail.com; 3Cátedras-CONACYT-Instituto Nacional de Electricidad y Energías Limpias, Morelos 62490, Mexico; lorena.magallon@ineel.mx

**Keywords:** microwave heating, green synthesis, Ru-based electrocatalyst, oxygen reduction

## Abstract

A simple and rapid synthesis of a CoFeRu-based electrocatalyst by a microwave-assisted method (using water as the microwave absorbing solvent) is reported in this work. Agglomerates with different sizes and shapes are observed by scanning electron microscopy technique. The energy dispersive X-ray spectroscopy shows a low atomic percentage of Co and similar atomic percentage of Fe and Ru. However, the X-ray diffraction exhibits only the presence of metallic Ru and Fe_2_O_3_ (hematite) phases. The oxygen reduction without and with 2 mol L^−1^ methanol is studied using the rotating disk electrode technique. The electrochemical kinetic parameters obtained are compared to a similar electrocatalyst reported in the literature, which was synthesized using a mixture of an organic solvent with DI water as the microwave absorbing solvent. An improvement on the activity of the electrocatalyst synthesized is observed, where high Tafel slopes are not observed. The electrocatalyst also showed tolerance to the presence of methanol during the oxygen reduction reaction.

## 1. Introduction

Hydrogen technology has generated great interest for its high efficiency and low pollutant emissions. In this way, polymeric electrolyte membrane fuel cells (PEMFCs) and direct methanol fuel cells (DMFCs) have been extensively investigated. However, these devices are limited by the low electrochemical activity at the cathode (where the reduction reaction takes place). In DMFCs, the methanol crossover effect is another critical step; this effect reduces the cathode efficiency by a competing electrochemical process, known as the mixed potential effect [1,2]. Moreover, when the methanol is oxidized on the active surfaces of Pt, an intermediate reaction produces carbon monoxide (−CO_ads_), which inactivates the active sites on the surface of the electrocatalyst [3,4]. To reduce these effects several Pt-free or Pt-alloy electrocatalysts have been reported in the literature [5,6,7,8,9,10,11,12].

In this way, cobalt-based electrocatalysts with attractive activity towards the oxygen reduction reaction (ORR) and tolerance to the presence of methanol have been studied [13,14,15,16,17]. Other kinds of electrocatalysts studied are those based on iron, where an iron loading of 4.7 wt% has shown the best activity towards the ORR, yielding a 1000 h lifetime as cathode in a fuel cell [18,19,20]. On the other hand, Alonso-Vante [21,22] has reported that Ru chalcogenides with Chevrel phase-type show significant catalytic activity towards the ORR in acidic media. Other Ru-M-based electrocatalysts (M = Se, Mo, Cr, Fe, Co, Pb) have shown activity and selectivity towards the four-electron mechanism for the oxygen reduction reaction [1,9,20,21,23,24,25,26]. On the other hand, the synthesis of carbon-decorated monometallic electrocatalysts (FeFe-N/C and CoCo-N/C) with attractive activity towards ORR has been reported; however, the FeCo-N/C carbon-decorated bimetallic electrocatalyst shows the best performance towards ORR [27]. Thus, when these metals are alloyed with Ru (RuFe and RuCo), their electrocatalytic activities are superior to the monometallic Ru electrocatalyst [28,29]. Another important feature of these bimetallic materials is that they show activity towards hydrogen oxidation reaction in an alkaline medium [29].

However, many of these electrocatalysts were synthesized using conventional heating methods, i.e., using an electric furnace or oil bath. The disadvantage of these methods is that there is no efficient heat transfer, since the first to heat up are the walls of the container vessel (reactor) and then the reactants either by convection or conduction. Therefore, long time periods (usually hours or even days) are needed to reach the target temperature. Besides that, organic or inorganic solvents as the reaction media are also used, i.e., they are not environmentally friendly methods.

On the other hand, high-speed chemical synthesis of inorganic nanostructured materials (such as metal, carbon, metal oxide and polymer nanocomposite materials) in liquid phase with microwaves has been used in recent years. This method involves effective heating of materials due to the dielectric heating effects of microwaves. In this way, the microwave-assisted synthesis is a fast and efficient technique that can be used to produce materials with greater reproducibility. The ability of a substance (a solvent for example) to transform electromagnetic energy into heat (at a given frequency and temperature) is determined by the loss factor (tanδ). In this way, solvents are classified as high (tanδ > 0.5), medium (tanδ 0.1–0.5) and low microwave absorbing (tanδ < 0.1). Therefore, the solvent plays a very important role for the microwave-assisted synthesis of inorganic nanostructures. Although water is classified as medium microwave absorbing (tanδ < 0.123), it becomes a very useful solvent for synthesis at high temperatures using the sealed-vessel technology [30,31,32,33,34]. Therefore, water is a very interesting solvent to be used at temperatures above its boiling point and high pressures (c.a. 60 bar) because it starts to behave like an organic solvent, and this property can be advantageous for synthesis of materials in aqueous media. In addition, water is a non-toxic and non-flammable solvent and is available in nature. On the other hand, when a chemical reaction is carried out in liquid solvents using a closed system it is called “solvothermal processing”, and when water is the only solvent, “hydrothermal processing”. This process allows to synthesize many materials in short times and at temperatures slightly below those required by conventional methods.

In this work, the effect of the solvent (deionized water) used for the synthesis of the electrocatalyst based on CoFeRu and its electroactivity towards the oxygen reduction reaction is reported. The electrocatalytic activity of the material is compared with a similar electrocatalyst synthesized by a microwave-assisted synthesis but using a mixture of an organic solvent and water [26]. This green chemical synthesis avoids the use of organic solvents and long times during the synthesis of electrocatalysts, and it is of great importance for a rapid development of this kind of material.

## 2. Materials and Methods

Electrocatalyst Synthesis. The CoFeRu (H_2_O) electrocatalyst was synthesized by mixing 67.5 mg of Co(NO_3_)_2_·6H_2_O (99.999%, Sigma-Aldric, St. Louis, MO, USA) with 50 mg of Ru_3_(CO)_12_ (99%, Sigma-Aldrich), 63.7 mg of FeSO_4_·7H_2_O (99%, Sigma-Aldrich) and 10 mL of DI-water (resistivity = 18.2 MΩ cm) and heated thermally using microwave irradiation in a microwave reactor (Synthos 3000, Anton Paar, Australia) at 180 °C during 30 min and at a maximum pressure of 60 bar. The product was centrifuged at 3000 rpm and washed with DI water and dried at room temperature.

Physical Characterization. The study of the chemical composition of the CoFeRu electrocatalyst was obtained by EDS (energy-dispersive X-ray spectroscopy, Hitachi High-Tech, Tokyo, Japan), using a Hitachi SU1510 microscope. The scanning electron microscopy (SEM, SEMTech Solutions, North Billerica, MA, USA) image was obtained using a JEOL JSM-7800F microscope (Tokyo, Japan). The structural characterization of the material was carried out by XRD (X-ray diffraction) analysis with a Rigaku DMAX-2200 diffractometer (Cu Kα1 radiation, 1.5406 Å, Rigaku Americas Corporation, The Woodlands, TX, USA). The X-ray diffraction pattern was obtained using Jade 6.5 software (MDI Material Data, Livermore, CA, USA). The Fourier transform infrared spectroscopy (FT-IR) was made by Bruker equipment, Tensor 27 model, using an ATR technique (Thermo Fisher Scientific, Waltham, MA, USA).

Electrochemical Characterization. An electrochemical cell with support for three electrodes, a potentiostat/galvanostat (Model AFCBP1, Pine Instrument, Durham, NC, USA) and Aftermath software (V 1.6.10513, Pine Instrument, Durham, NC, USA) were used for electrochemical measurements and recording of experiments. The working, reference and auxiliary electrodes were glassy carbon disk, mercury sulfate (Hg/Hg_2_SO_4_/0.5 mol L^−1^ H_2_SO_4_) (0.680 V/NHE) and carbon rod, respectively; 0.5 mol L^−1^ H_2_SO_4_ (98%, J.T. Baker, Fisher Scientific, Madrid, Spain) was used as the electrolyte. The method used for the preparation of the CoFeRu/C electrocatalytic ink was as follows: first, 0.6 mg of CoFeRu was mixed with 1.4 mg of Vulcan XC-72R (Fuel Cell) and 15 μL of 5% Nafion/isopropanol solution (ElectroChem) in an ultrasonic bath (Branson 1510). Then, 5 μL of the electrocatalytic ink was deposited on the surface of the glassy carbon disk electrode (geometrical surface area = 0.1963 cm^2^, mirror polished with 0.05 μm alumina, Pine instruments) and finally dried at room temperature. The final loadings of the CoFeRu electrocatalyst and Nafion^®^ were 1020 μg cm^−2^ and 1200 μg cm^−2^, respectively. These loadings are similar to those used for Ru-Fe and CoFeRu materials described in the literature. The electrolyte was bubbled with nitrogen (Infra; UHP), and then cyclic voltammetry (CV) was carried out between 0 and 0.98 V/NHE (the sweep rate is 20 mV s^−1^) until reproducible voltammograms were obtained. After that, the electrolyte was bubbled with oxygen (Infra; UHP) until no changes in the cell potential (open circuit potential, OCP) were observed (~30 min), then the oxygen reduction was carried out by linear sweep voltammetry (LSV) from the OCP-0 V/NHE (the sweep rate is 5 mV s^−1^), under rotating conditions (100, 200, 400, 600 and 900 rpm) using the rotating disk electrode technique (RDE). CV and LSV studies were also obtained with 2 mol L^−1^ methanol (CH_3_OH 99.9%, J.T. Baker) present in the electrolyte, at the same conditions mentioned above.

## 3. Results and Discussion

### 3.1. Physica and Structural Characterization

The scanning electron microscopy (SEM) image of the CoFeRu electrocatalyst synthesized in this work at 43,000× magnification is shown in Figure 1. It is observed as cloud-like aggregates with irregular shapes and sizes and a material size greater than 100 nm. It is not possible to identify a specific type or shape of particle.

The chemical composition (at.%, atomic percentage) of the electrocatalyst synthesized is shown in Table 1, which is compared with the previously reported composition of CoFeRu synthesized using a mixture of ETG/H_2_O (ethylene-glycol/DI-water). In general, it is observed that the material synthesized in water, CoFeRu (H_2_O), showed a lower at.% of Co and Ru than CoFeRu (ETG/H_2_O). Considering the at.% of Fe, O and C in CoFeRu (H_2_O), these values can be related to the presence of Fe_2_O_3_ and CO formulas. The presence of CO was confirmed with FT-IR results shown in Figure 3. For the CoFeRu (ETG/H_2_O) electrocatalyst, the presence of carbonyl groups in its chemical composition was not reported. This new composition of the electrocatalyst could be attributed to the use of water as the microwave absorbing solvent, which favors the formation of iron oxide. A similar behavior was observed for the Ru-Fe electrocatalyst reported by Su [35].

Figure 2 shows the XRD pattern of the unsupported CoFeRu electrocatalyst. Defined peaks/(crystallographic planes) were observed at 24.06°/(1 0 2), 33.02°/(1 0 4), 35.62°/(2 −1 0), 49.34°/(2 0 4), 53.86°/(2 −1 6) 63.96°/(3 0 0) and 84.94°/(4 −1 4) associated to the Fe_2_O_3_ phase [hematite, R-3c (167)] showing a crystallite size of 22 nm ± 6 nm. The co-existence of a crystalline Ru phase [ruthenium, P63/mmc (194)] was observed at 38.40°/(1 0 0), 42.20°/(0 0 2), 44.02°/(1 0 1), 58.54°/(1 0 2), 69.48°/(2 −1 0), 78.50°/(1 0 3) and 86.08°/(2 −1 2), showing a crystallite size of 9 nm ± 2 nm. These crystallite sizes were calculated by the Scherer formula; however, due to the irregular morphology and the size of the aggregates the estimations presented a wide standard deviation. Additionally, the XRD pattern presented a low-angle area associated to an amorphous phase. The Ru and Fe_2_O_3_ phases were identified by JCPDS card No. 06-0663 and JCPDS card No. 33-0664, respectively. No cobalt-associated signals were observed in the XRD pattern, possibly due to the low Co at.% in the material.

The FT-IR spectra of the CoFeRu electrocatalyst is shown in Figure 3, along with the spectra of the precursors Ru_3_(CO)_12_, Co(NO_3_)_2_·6H_2_O and FeSO_4_·7H_2_O. Very broad O–H stretching vibration bands around 3700–3000 cm^−1^ were observed for Co and Fe salts [36], N–O asymmetric bond vibration bands at 1640 and 1382 cm^−1^ were also observed for Co salt [37,38], while Fe salt showed two peaks between 1100 and 1700 cm^−1^ corresponding to S=O mode. Ru_3_(CO)_12_ showed the presence of a terminal carbonyl stretching vibration region [M-(CO)] and a group of bands associated to carbonyl deformation modes at 1800–2200 cm^−1^ and around 570 cm^−1^, respectively [39]. The CoFeRu electrocatalyst presented a signal around 1870 cm^−1^, which can be associated either with a terminal carbonyl [μ_1_-(CO)] or a bridging carbonyl (μ_2_-CO), given the fact that the frequency of the material was in the limit between both carbonyl frequencies. This means that Ru_3_(CO)_12_ did not undergo complete decarbonylation.

### 3.2. Electrochemical Studies

The cyclic voltammograms of CoFeRu/C with and without methanol are shown in Figure 4. The reduction of a ruthenium oxide thin film formed during the anodic sweep [40] was observed between 0.1 and 0.4 V/NHE as a cathodic peak, while hydrogen adsorption/desorption regions (between 0 and 0.1 V/NHE) and iron redox couple signs (between 0.6 and 0.8 V/NHE) were also shown by the material [35]. No peaks associated with methanol oxidation were observed, i.e., the electrocatalyst did not show activity towards this reaction.

The disc current densities under rotating conditions of CoFeRu/C for the oxygen reduction with and without methanol is shown in Figure 5a. The RDE technique allows to distinguish between mass transport and reaction kinetics. In this way, from the onset potential to the head of the plateaus the mixed, kinetic and diffusion-limited regions are observed; while at more cathodic potentials, where the limiting current plateaus are reached, the diffusion limited region is observed. The presence of methanol during the oxygen reduction reaction slightly decreases the current density. This effect could be due to the adsorption of methanol on the electrode surface. However, the open circuit potential is not affected by the presence of methanol. Figure 5b shows the oxygen reduction reaction studies at 900 rpm for CoFeRu (H_2_O) and CoFeRu (H_2_O/ETG) and Ru-Fe reported in the literature as comparison. In general, the CoFeRu (H_2_O) electrocatalyst showed higher current densities and open circuit potential (OCP, Table 2) values than Ru-Fe and CoFeRu reported in the literature, even with 2 mol L^−1^ methanol.

The Koutecký–Levich analysis allows to obtain critical kinetic parameters for an electrochemical reaction. The KL equation (Equation (1)) relates the sum of the reciprocals of the kinetic and the diffusion limited currents, i_k_ and i_d_, respectively, as shown below:(1)$$\frac{1}{i}=\frac{1}{i_{k}}+\frac{1}{i_{d}}$$
which is calculated based on the Levich equation (Equation (2)),
(2)$$i_{d}=200\;{\mathrm{nFAC}}_{O_{2}}D_{O_{2}}^{2/3}\nu^{-1/6}\omega^{1/2}$$
where [31] A = geometric area of the glassy carbon disk (0.1963 cm^2^),F = Faraday’s constant (96,485 C mol^−1^),𝜐 = kinematic viscosity of the electrolyte (0.01 cm^2^ s^−1^),$D_{O_{2}}$ = diffusion coefficient of O_2_ in the electrolyte (1.40 × 10^−5^ cm^2^ s^−1^),$C_{O_{2}}$ = concentration of O_2_ in the electrolyte (1.1 × 10^−6^ mol cm^−3^),ω = electrode rotation velocity (rpm),n = number of electrons involved in the oxygen reduction reaction,200 = constant used when ω is expressed in rpm and the current disk (i_d_) in mA.

In this way, Equation (1) can be rewritten as
(3)$$\frac{1}{i}=\frac{1}{i_{k}}+\frac{1}{200\;{\mathrm{nFAC}}_{O_{2}}D_{O_{2}}^{2/3}\nu^{-1/6}\omega^{1/2}}$$
or as
(4)$$\frac{1}{i}=\frac{1}{i_{k}}+\frac{\omega^{-1/2}}{B_{L}}=\frac{1}{i_{k}}+B_{\mathrm{KL}}\cdot \omega^{-1/2}$$
where B_L_ (= 200 ${\mathrm{nFAC}}_{O_{2}}D_{O_{2}}^{2/3}\nu^{-1/6}$) and B_KL_ (= 1/ B_L_) are the Levich and Koutecký–Levich slopes, respectively.

The Koutecký–Levich plots (1/i vs. 1/ω^1/2^) are fitted using linear regression to calculate the KL slopes and estimate the number of electrons involved during the ORR [31]. In this way, the theoretical (2 and 4 e^−^) and experimental Koutecký–Levich plots (at 0.24 V/NHE) without and with 2 mol L^−1^ methanol are shown in Figure 6. Clearly, the experimental lines in Figure 6 come near to the theoretical 4 e^−^ transfer, i.e., the reaction mechanism for the oxygen reduction involves a four-electron pathway directly to H_2_O [41,42,43,44]. This behavior was not altered even by the presence of methanol.

The Butler–Volmer equation (Equation (5)) is the standard approach in macroscopic modeling of electrochemical kinetics in porous electrodes. When mass transfer is not considered, the Butler–Volmer equation can be expressed as a general current-overpotential equation [45].
(5)$$i_{k}=i_{o}\left[e^{-\frac{\alpha F}{\mathrm{RT}}\eta }-e^{\frac{\left(1-\alpha \right)F}{\mathrm{RT}}\eta }\right]$$

For large negative overpotentials (η = E − E_eq_), the Butler–Volmer equation becomes
(6)$$i_{k}=i_{o}e^{-\frac{\alpha F}{\mathrm{RT}}\left(E-E_{\mathrm{eq}}\right)}$$

Equation (6) can be rewritten as
(7)$$i_{k}={\mathrm{Ae}}^{-\frac{\alpha F}{\mathrm{RT}}E}$$
where
(8)$$A=\frac{i_{0}}{e^{-\frac{\alpha F}{\mathrm{RT}}E_{\mathrm{eq}}}}$$

Expressing Equation (7) in its logarithmic form, it becomes
(9)$${\mathrm{logi}}_{k}=\mathrm{logA}-\frac{\alpha F}{2.3\mathrm{RT}}E$$

In this way, the Tafel plots corrected by mass-transport (log i_k_ vs. E, Figure 7) for the ORR without and with methanol can be obtained from Equation (9).

In order to get the kinetic current (i_k_) from the Koutecký–Levich equation as shown in Equation (10), the potential–current curves are corrected by a previously described procedure [46,47,48].
(10)$$i_{k}=\frac{i\cdot i_{d}}{i-i_{d}}$$

Table 2 summarizes the electrokinetic parameters obtained from Figure 7, i.e., Tafel slope, charge transfer coefficient and the exchange current density. These parameters are compared with CoFeRu and Ru-Fe electrocatalysts reported in the literature.

Platinum is one of the most active electrocatalysts used for the oxygen reduction reaction, and for this electrocatalyst two Tafel slopes were observed: 60 mV decade^−1^ and 120–160 mV decade^−1^ at low current density (lcd) and high current density (hcd) regions, respectively. At the lcd region the adsorbed oxygen species showed the fast initial electron transfer step followed by the rate-determining chemical step, i.e., adsorption under the Temkin adsorption conditions; while at the hcd region, the initial electron transfer was the rate-determining step under the Langmuir adsorption conditions [49]. On the other hand, Tafel slopes above 120 mV decade^−1^ are often observed in reactions through some adsorbed ruthenium oxide layers on the surface, as observed with the reported CoFeRu and Ru electrocatalysts, whose Tafel slope values were above 200 mV decade^−1^. Therefore, Ru oxides of various monolayer thickness could lead to an increase in the Tafel slope in these materials, which could be caused by a different reaction mechanism for oxygen absorption, making the analysis of the reaction mechanism more difficult [50]. On the other hand, the expected Tafel slope (120 mV decade^−1^) at hcd was observed for the CoFeRu (H_2_O) electrocatalyst. This behavior could be due to the fact that only a very thin layer of ruthenium oxide formed on the surface, which was not enough to affect the oxygen adsorption mechanism.

According to the relationship between the Tafel slope (b) and the charge transfer coefficient (α) (b = 2.3RT/αF), low α values are expected at high Tafel slope values, as observed with CoFeRu and Ru-Fe electrocatalysts reported in the literature. However, this α value for the CoFeRu (H_2_O) electrocatalyst was close to 0.5, as expected for Tafel slopes near 120 mV decade^−1^.

Finally, the exchange current density (j_o_) is directly related to the rate constant and represents the measure of the charge transfer rate in an electrochemical reaction at equilibrium [41], i.e., the higher this value is, the easier the reaction is to initiate. This electrokinetic parameter for the ORR on Pt/C electrocatalyst was 3.9 × 10^−2^ mA cm^−2^ at the hcd region [49], i.e., this electrochemical parameter obtained for the CoFeRu (H_2_O) electrocatalyst was only three orders of magnitude lower than Pt/C.

## 4. Conclusions

A microwave thermal heating process has been employed to synthesize a CoFeRu electrocatalyst, using only deionized water as the microwave absorbing solvent (green chemical synthesis), i.e., the use of organic solvents and long synthesis times are avoided. Although the structural characterization revealed only the presence of three phases—Ru, Fe_2_O_3_ and an amorphous phase—the chemical composition analysis showed the presence of Co, which was not detected by X-ray diffraction due to its low at.%. SEM studies showed the formation of agglomerates with different sizes and shapes. The electrochemical performance of CoFeRu was improved compared to that synthesized in an ethylene-glycol/DI-water mixture, favoring the formation of Fe_2_O_3_ rather than a ruthenium oxide. The electrocatalyst showed activity towards the ORR even in the presence of up to 2 mol L^−1^ methanol. Therefore, this electrocatalyst can be proposed as cathode in a PEMFC or DMFC.

## Figures and Tables

**Figure 1 materials-14-01662-f001:**
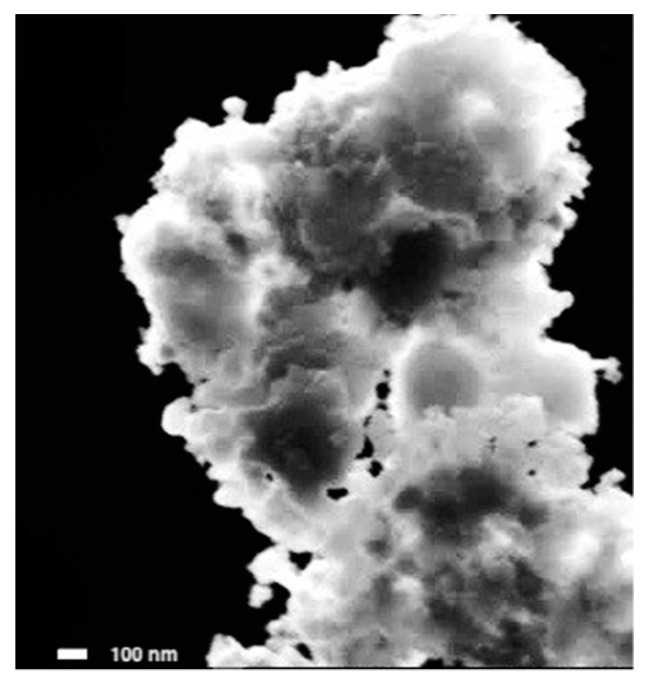
Scanning electron micrographs of CoFeRu at 43,000×.

**Figure 2 materials-14-01662-f002:**
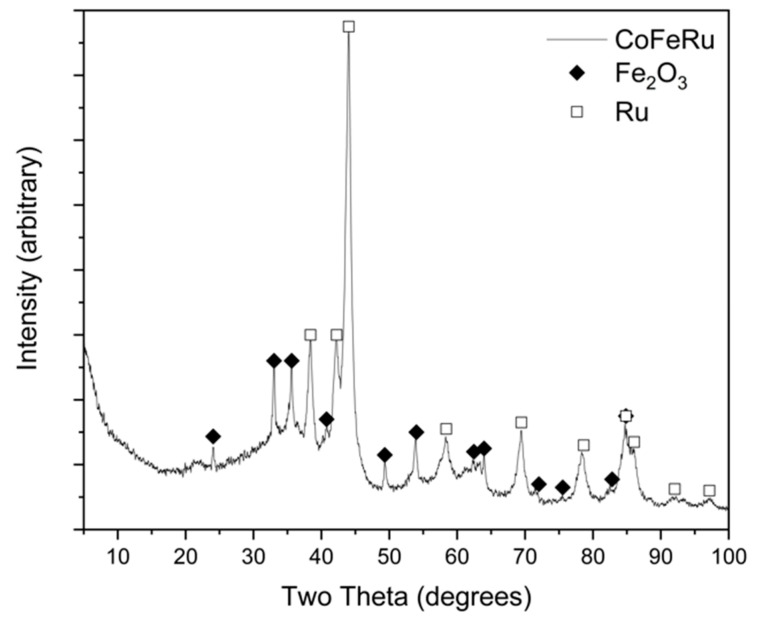
XRD pattern of unsupported CoFeRu electrocatalyst.

**Figure 3 materials-14-01662-f003:**
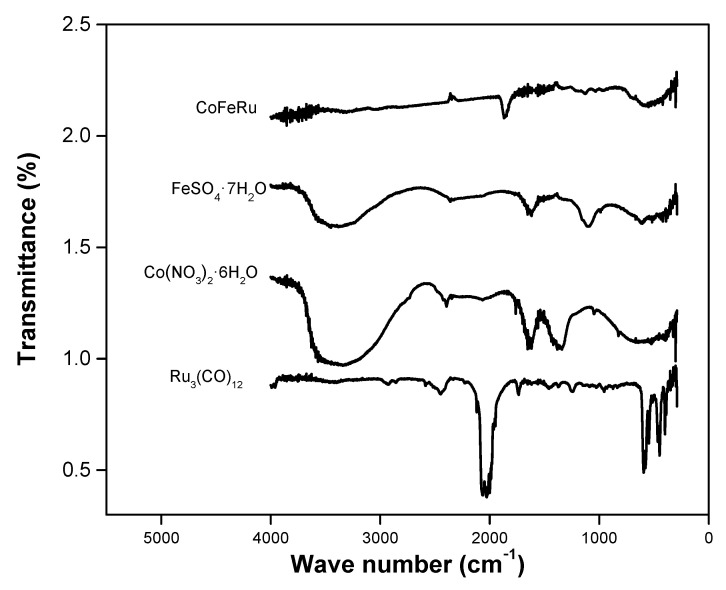
FT-IR spectra of the unsupported CoFeRu electrocatalyst and precursors.

**Figure 4 materials-14-01662-f004:**
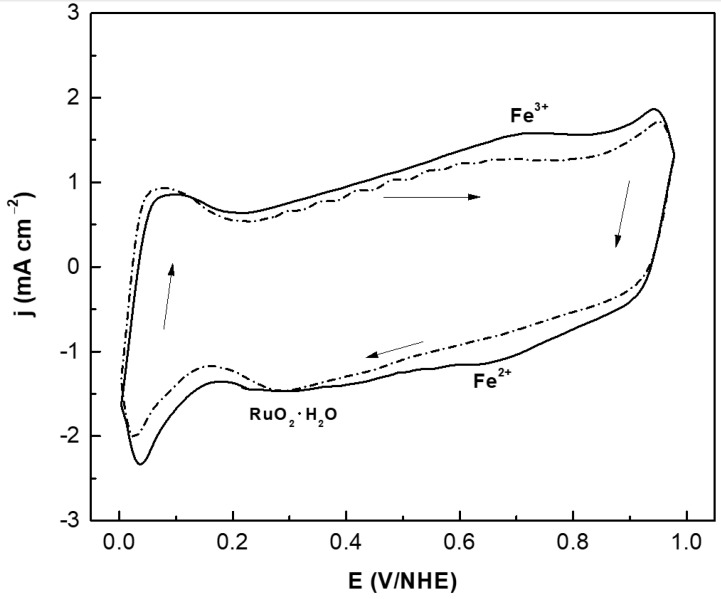
Cyclic voltammograms of CoFeRu/C without (solid line) and with (dash line) 2 mol L^−1^ methanol. The sweep rate was 20 mV s^−1^ and as electrolyte 0.5 mol L^−1^ H_2_SO_4_.

**Figure 5 materials-14-01662-f005:**
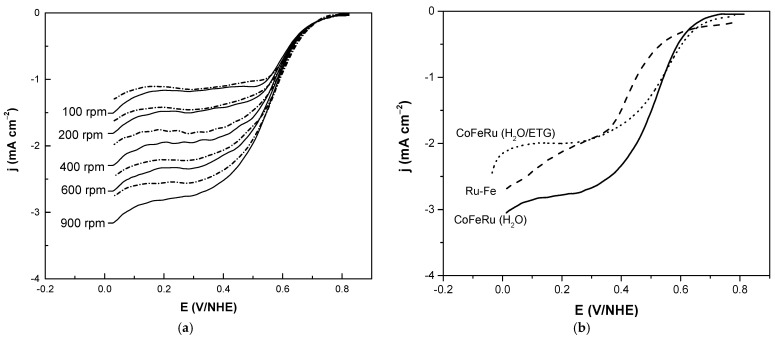
Linear sweep voltammograms for the oxygen reduction of (**a**) CoFeRu/C electrocatalyst without (solid line) and with (dash line) 2 mol L^−1^ methanol; (**b**) CoFeRu (H_2_O), CoFeRu (H_2_O/ETG) and Ru-Fe reported in the literature at 900 rpm. The sweep rate was 5 mV s^−1^ and as electrolyte 0.5 mol L^−1^ H_2_SO_4_.

**Figure 6 materials-14-01662-f006:**
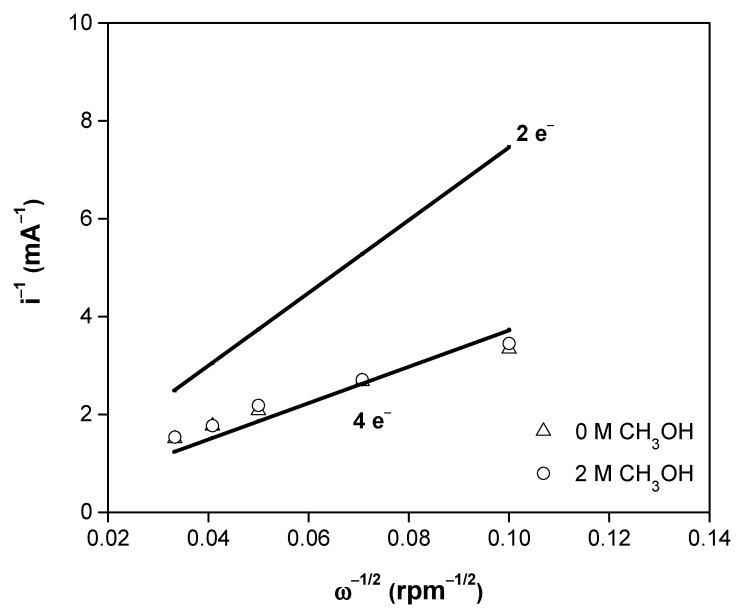
Theoretical (2 and 4 e^−^, solid lines) and experimental Koutecký–Levich plots of CoFeRu/C without (∆) and with (ο) 2 mol L^−1^ methanol.

**Figure 7 materials-14-01662-f007:**
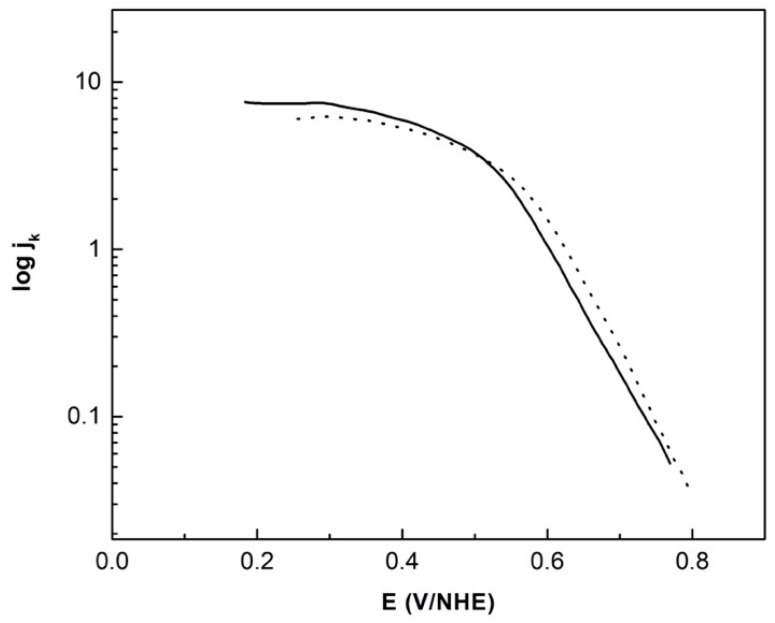
Tafel plots (corrected by mass-transfer) of CoFeRu/C without (−) and with (--) 2 mol L^−1^ methanol.

**Table 1 materials-14-01662-t001:** EDS of CoFeRu electrocatalysts synthesized in deionized water and in a mixture of ethylene-glycol/deionized-water (ETG/H_2_O).

Electrocatalyst	at.%				
	Co	Fe	Ru	O	C
CoFeRu (H_2_O)	0.47	17.52	13.30	47.47	21.24
CoFeRu (H_2_O/ETG) [26]	1.70	1.70	46.60	33.00	17.00

**Table 2 materials-14-01662-t002:** Electrokinetic parameters and open circuit potentials of CoFeRu electrocatalyst synthesized using deionized water, and CoFeRu and Ru-Fe electrocatalysts reported in the literature.

Electrocatalyst	Methanol (mol L^−1^)	Open Circuit Potential (V/NHE)	Tafel Slope (V dec^−^^1^)	Charge Transfer Coefficient	Exchange Current Density (mA cm^−^^2^)
CoFeRu (H_2_O)	0	0.811 (0.002)	0.129 (0.001)	0.420 (0.004)	1.36 (0.09) × 10^−5^
	2	0.823 (0.0006)	0.126 (0.002)	0.430 (0.007)	1.55 (0.19) × 10^−5^
CoFeRu (ETG/H_2_O) [26]	0	0.782	0.193	0.306	2.49 × 10^−4^
	2	0.780	0.203	0.306	3.20 × 10^−4^
Ru-Fe [35]	0	0.772	0.267	0.22	1.7 × 10^−3^
	2	0.774	0.270	0.22	2.0 × 10^−3^

## Data Availability

Data sharing is not applicable to this article.

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
