# Peer review of "Green Microwave-Assisted Synthesis of CoFeRu-Based Electrocatalyst for the Oxygen Reduction Reaction"

_materials, 2021, doi:10.3390/ma14071662_

Round 1

Reviewer 1 Report

The manuscript reports a microwave-assisted method to prepare CoFeRu catalyst for the oxygen reduction reaction. The material has been characterized by SEM, XRD, FTIR, EC scanning. Generally, the work is interesting to the readers of this journal and it can be considered for being accepted after solving the following significant issues. 

(1)  the motivation to hybridized Co, Fe, Ru is not clear. Multi-metallic ORR catalysts should be introduced in the introduction. See Small, 2017, 13, 1604103 ;  Applied Catalysis B: Environmental 282 (2021) 1195932

(2) What is the function of Co, Fe, Ru in the composite, control experiment should be conducted to compare CoFeRu with CoFe, CoRu, FeRu in terms of the ORR activity.  Otherwise, there is no point in preparing such a complicated composition. 

(3) In addition to Co, Fe, Ru, O, C, does any other element exist in the composite?

(4) It seems the author prepared a hybrid catalyst composed of several components, what is the structure of Co? 

Author Response

We thank the reviewer for his observations, comments and questions. They have been very helpful in clarifying some points. These remarks and comments have been taken into account and corresponding changes have been made in the manuscript. These modifications are highlighted in yellow in the main text.

On the other hand, the English language and style has been revised and some parts of the manuscript have been rewritten as suggested also by the editor.

The manuscript reports a microwave-assisted method to prepare CoFeRu catalyst for the oxygen reduction reaction. The material has been characterized by SEM, XRD, FTIR, EC scanning. Generally, the work is interesting to the readers of this journal and it can be considered for being accepted after solving the following significant issues.

(1) The motivation to hybridized Co, Fe, Ru is not clear. Multi-metallic ORR catalysts should be introduced in the introduction. See Small, 2017, 13, 1604103 ; Applied Catalysis B: Environmental 282 (2021) 1195932

Dear Reviewer, thank you for your comment and suggestion. Although the manuscript indicates that different electrocatalysts based on Ru, Fe and Co with attractive electrocatalytic activity towards oxygen reduction reaction have been reported, we have highlighted in the manuscript the synthesis of multi-metallic electrocatalysts based on these materials (FeFeN/C, CoCoN/C, FeCo/N, RuFe and RuCo) (new references in the manuscript: 27-29).

In summary, the main reason that motivated us to further study the multi-metallic material CoFeRu was that the activity of the monometallic Ru electrocatalyst is enhanced when it is synthesized in its bimetallic form with Fe or Co, i.e., Ru-Co or Ru-Fe. Therefore, it was proposed to synthesize an electrocatalyst with the three Ru-Co-Fe metals, and to study the effect they have together on the oxygen reduction reaction. Which was previously synthesized but using a mixture of an organic solvent with deionized water. In this way, this work proposes the synthesis of CoFeRu avoiding the use of organic solvents, i.e., using a green chemistry synthesis. Observing that the new material showed in general a better electrochemical performance than previously reported.

(2) What is the function of Co, Fe, Ru in the composite, control experiment should be conducted to compare CoFeRu with CoFe, CoRu, FeRu in terms of the ORR activity. Otherwise, there is no point in preparing such a complicated composition.

Dear Reviewer, thank you for your comment. According to the literature (28 and 29 in the manuscript), the improvement in the performance of Ru-Co-Fe based electrocatalysts is mainly due to electronic and/or the ensemble effects. For example, it has been found that Co atoms are incorporated into the Ru lattice in RuCo electrocatalyst and this effect could be occurring with the CoFeRu material reported in this manuscript, and for that reason no Co-associated crystalline phase is observed by x-ray studies.

On the other hand, as mentioned above in point (1), RuCo and RuFe electrocatalyst show better performance than the monometallic Ru. And for that reason it was decided to study the effect of having the three metals in the electrocatalyst.

(3) In addition to Co, Fe, Ru, O, C, does any other element exist in the composite?

Dear Reviewer, thank you for your comment. According to the region selected for ESD studies, these are the main components in the compound.

(4) It seems the author prepared a hybrid catalyst composed of several components, what is the structure of Co?

Dear reviewer, thank you for your comment. Unfortunately, with the information we have it has not been possible to determine the structure of the Co. However, it has been reported in the literature that electrocatalysts based on Co, CoFe and RuCo have generally shown a hexagonal alloy phase with small amounts of metal oxides on the surface, both in the monometallic and bimetallic form of Co (references 28 and 29).

Reviewer 2 Report

Sandoval et al report a new synthetic method for a CoFeRu electrocatalyst aiming at oxygen reduction.

They provide spectral as well electrochemical characterization of two types of CoFeRu catalyst materials one involving water and one ethylene glycol. They finally evaluate the efficiency of the electrocatalyst and compare the results to another published CoFeRu catalyst. The work is sound and the presentation of the results is done in a nice way allowing the reader to follow all the steps of the experimental work pursued. I have a major concern only on one issue which in my opinion should be addressed:

Is the synthesis reproducible enough to ensure a good efficiency for oxygen reduction from batch to batch? The differences with other published similar electrocatalysts is not huge and it is questionable if this corresponds to a reproducible effect when using another batch derived through the same  methodology. The authors in my opinion should address this by indicating the results of various batches and finally present the average efficiency of their electrocatalyst.

Additionally, I was wondering about the reusability of this material. Are there any data on regeneration of the catalyst and potentially  efficiency dependence on number of catalytic cycles?

Minor issues: some typos need to be corrected e.g. physical instead of physical etc.

Author Response

"PLEASE SEE THE ATTACHMENT"

Round 2

Reviewer 1 Report

The authors have addressed all the concerns of the reviewer. And the present version can be published as it is. 

Reviewer 2 Report

The authors have addressed all the issues pointed out during the first round of revisions. I believe the manuscript is suitable for publication in its current form.